# Validation of Copy Number Variants Detection from Pregnant Plasma Using Low-Pass Whole-Genome Sequencing in Noninvasive Prenatal Testing-Like Settings

**DOI:** 10.3390/diagnostics10080569

**Published:** 2020-08-08

**Authors:** Michaela Hyblova, Maria Harsanyova, Diana Nikulenkov-Grochova, Jitka Kadlecova, Marcel Kucharik, Jaroslav Budis, Gabriel Minarik

**Affiliations:** 1Trisomy Test s.r.o., Ilkovičova 8, 841 04 Bratislava, Slovakia; gabriel.minarik@medirex.sk; 2Medirex a.s., Galvaniho 17/C, 821 06 Bratislava, Slovakia; 3Geneton s.r.o., Galvaniho 7, 821 06 Bratislava, Slovakia; maria.harsanyova@geneton.sk (M.H.); marcel.kucharik@geneton.sk (M.K.); jaroslav.budis@geneton.sk (J.B.); 4Cytogenetic laboratory Brno s.r.o., Veveri 39, 602 00 Brno, Czech Republic; diana.groch@gmail.com (D.N.-G.); jkd@centrum.cz (J.K.)

**Keywords:** CNV detection, noninvasive prenatal testing (NIPT), low-pass whole-genome sequencing

## Abstract

Detection of copy number variants as an integral part of noninvasive prenatal testing is increasingly used in clinical practice worldwide. We performed validation on plasma samples from 34 pregnant women with known aberrations using cell-free DNA sequencing to evaluate the sensitivity for copy number variants (CNV) detection using an in-house CNV fraction-based detection algorithm. The sensitivity for CNVs smaller than 3 megabases (Mb), larger than 3Mb, and overall was 78.57%, 100%, and 90.6%, respectively. Regarding the fetal fraction, detection sensitivity in the group with a fetal fraction of less than 10% was 57.14%, whereas there was 100% sensitivity in the group with fetal fraction exceeding 10%. The assay is also capable of indicating whether the origin of an aberration is exclusively fetal or fetomaternal/maternal. This validation demonstrated that a CNV fraction-based algorithm was applicable and feasible in clinical settings as a supplement to testing for common trisomies 21, 18, and 13.

## 1. Introduction

The application of noninvasive prenatal testing (NIPT) has increased dramatically from fetal sex determination to whole fetal genome sequencing. This is a consequence of rapidly evolving Next Generation Sequencing (NGS) techniques that have brought revolutionary changes in many areas, including prenatal diagnosis. Detection of common trisomies 21, 18, and 13 has become routine, as the sensitivity has reached over 99% for worldwide NIPT. The applied approaches include either whole-genome or a targeted approach [1,2] as the application of NIPT is rapidly expanding beyond whole chromosomes, including quantitative sub-chromosomal aberrations, known as copy number variants (CNVs). Unlike common trisomies, the incidence of CNVs is independent of maternal age [3], and prenatal detection of pathogenic CNVs would, therefore, benefit all pregnant women. However, the widespread implementation of CNV detection in standard prenatal screening is limited, as the detection sensitivity is still under investigation.

The recently published data from NIPT screening showed that 4.2% of individuals in the Central European region carry a CNV ≥ 600 kilobases (kb) [4], suggesting a higher frequency than commonly reported. A study by Ke et al. on 14,235 NIPT cases found an incidence rate of CNVs in the general pregnant population of 1.69%. However, small aberrations (< 3 megabases (Mb)) were not detected or taken into account in the overall calculation [5].

Detection of small CNVs by NIPT remains a challenge as the proportion of cell-free fetal DNA bearing the imbalanced chromosomal segments is limited [6]. On average, cell-free fetal DNA represents only 10% of cell-free DNA in maternal plasma (fetal fraction), while the remaining 90% is of maternal origin. The most critical parameters for CNV detection in the NIPT environment include fetal fraction, size of the aberration, sequencing depth, and the biological variability of the region. A sufficient fetal fraction, which comprises the proportion of fetal DNA against the predominant background of maternal DNA, is vital in the detection of an aberration inversely proportional to its size. Therefore, a higher fetal fraction is imperative for a smaller CNV.

It is much easier to detect a larger event with a higher fetal fraction. On the other hand, the read counts expressed as uniquely mapped reads (UMR) represent the depth of genomic coverage in a given area. Higher read counts are positively correlated with detection sensitivity. Moreover, some genomic regions are inherently more variable (regions with repetitions, lower mapping ability, etc.), causing lower sensitivity in CNV detection. First, a proof-of-principle validation was undertaken on artificial samples. These samples represented a mix of non-pregnant female plasma and well-defined microaberration-positive DNA simulating fetal fractions in serial dilutions from 5% to 17.5% [7]. A very deep sequencing strategy from 144 million (M) up to 750 M sequencing tags was also reported in several previous studies [8,9,10]. However, this approach is costly and, therefore, inapplicable and inadaptable for routine clinical use. As a result, a great deal of effort has been dedicated to the improvement of bioinformatic pipelines to address this problem. Yu et al. employed a Non-Invasively Prenatal Sub-Chromosomal Copy number variation Detection (NIPSCCD) algorithm and evaluated its performance on more than 20,000 clinical samples. The team achieved more than 90% sensitivity for CNVs of 5 Mb–10 Mb and 100% sensitivity for those ≥ 10 Mb with an average of 7.5 M sequencing tags [11]. Straver et al. showed that 71.8% of CNVs ranging from 0.52 Mb to 84 Mb could be detected with as little as 3.5 M tags using the algorithm titled WISECONDOR (WIthinSamplE COpy Number aberration DetectOR), but with lower sensitivity (41.2% between 1 Mb and 5 Mb) [12].

We extended our previous in silico study using artificially created data sets and laboratory analyses on artificial DNA samples, with known microdeletions. The predicted sensitivity reached 79.3% for samples with fetal fraction ≥ 10% and a 20 M read count, and it further increased to 98.4% when focusing only on CNVs larger than 3 Mb [13].

The objective was to evaluate the clinical feasibility of NIPT to detect CNVs of different sizes with various fetal fractions on real pregnant plasma samples using a CNV fraction-based algorithm.

## 2. Materials and Methods

### 2.1. Clinical Samples

In this validation experiment, plasma DNA from pregnant women with a singleton pregnancy subjected to an invasive procedure due to ultrasound findings was used. All the participating women were recruited at a prenatal diagnostic center in Brno, Czech Republic, and Bratislava, Slovakia, between July 2015 and November 2018. All the pregnant women gave their informed consent for inclusion before participation. The study was conducted in accordance with the Declaration of Helsinki, and the protocol was approved on 30 June 2015 by the Ethics Committee of the Bratislava Self-Governing Region (03899/2015/HF). The demographic information on the samples in the cohort is summarized in Table 1.

Information on the size and type of CNVs in all samples was blinded until the procedure was completed, including CNV analysis. Finally, the only analyses included in the evaluation were those assumed not to share any aberration with the maternal genome. All CNV results were compared with the results of chromosomal microarray (CMA) (Agilent, ISCA 8x60K v2).

### 2.2. Blood Draw and Isolation

Maternal peripheral blood (10 mL) was drawn in EDTA or STRECK tubes prior to the invasive procedure (amniocentesis). Written consent was obtained from all participants. Blood samples were inverted several times (vigorous shaking cannot be used) after collection, stored in a chilled environment (4–10 °C) for EDTA (room temperature for STRECK tubes), and transported to the laboratory within 36 h. Maternal blood plasma was separated using centrifugation at 1,600× *g* for 10 min at 4 °C, with a subsequent centrifugation step at 16,000 × *g* for 10 min. Cell-free DNA (cfDNA) was extracted from 700 µl plasma using the QIAamp DNA Blood Mini Kit (Qiagen, Hilden, Germany) according to the manufacturer’s protocol and stored at −20 °C to be subject to further analysis. The final concentration of plasma DNA (expected to be of cell-free origin) was measured using a fluorometric method and did not exceed 0.3 ng/µL, which was strongly indicative of maternal nuclear cell DNA contamination due to non-compliance of pre-analytical conditions.

### 2.3. Library Construction and Sequencing

For adapter-ligated DNA library construction, a TruSeq Nano (Illumina, San Diego, CA, USA) kit with an in-house optimized protocol was used as previously published [14,15]. The starting input of cfDNA was typically less than 5 ng (~ 0.1 ng/µl). Low coverage sequencing (0.3 ×) was performed on Illumina NextSeq 500/550 platform (Illumina, San Diego, CA, USA) with paired-end setting 2 × 35 using High Output Sequencing Kit v2.5. Library quantity and quality were measured by fluorometric assay on Qubit 2.0 (ds DNA HS Assay Kit, Life Technologies, Eugene, Oregon, USA), and fragment analysis was performed on 2100 Bioanalyzer (High Sensitivity DNA Kit, Agilent Technologies, Waldbronn, Germany). We targeted 20 M UMR per sample; however, none of the analyses were excluded due to lower read counts.

### 2.4. Mapping and CNV Analysis

Sequencing reads were aligned to the reference genome hg19 (NCBI build 37) using the Bowtie2 algorithm [16]. Fetal fractions were calculated from the Y chromosome [17] in pregnancies carrying a male fetus and using the “combined” method as previously published [18] for female fetuses. To identify microaberrations, we grouped the reads per bin (20 kb bin size). We then employed a two-step normalization: LOESS-based correction [19] and PCA normalization to remove higher-order population artifacts on autosomes [7]. Finally, the signal was split into regions with equal level signals using the circular binary segmentation algorithm from the R package DNAcopy [20]. For each detected CNV, the corresponding CNV fraction value was calculated as follows: Firstly, we labeled the number of reads mapped on the detected CNV as rc and the average number of reads mapped on a similarly long region as rm. Then, the CNV fraction was computed based on the following formula: ffcnv = 2 × abs(rc−rm)/rm. The resulting data was visualized using an in-house tool (Figure 1). These figures are automatically generated for each chromosome, including X and Y.

## 3. Results

In total, we analyzed 34 plasma samples with confirmed CNVs that were blinded for their CNV status regarding type, size, and position. Among these, we identified 40 notable events. Of the total 40 CNVs, 29 were considered fetal and were further analyzed for technical validation based on the CNV fraction counted for the given aberration. Among the further analyzed CNVs, there were 15 duplications and 14 deletions. The CNVs ranged in size from 1.12 Mb up to 90 Mb. Regarding the deletions, the results were concordant with chromosomal microarray (CMA) in 14 out of 17 cases (82.4%). When it comes to duplications, 15 of 15 (100%) were detected correctly. Overall sensitivity reached 90.62% (29/32) regardless of size, fetal fraction, or read count. The number of CNVs smaller than 3 Mb was 14, and detection sensitivity in this group reached 78.57% (11/14). In the group with an aberration size larger than 3 Mb, detection sensitivity was 100% (18/18). The median fetal fraction was 14.59% (6.82–30.10%), and the median of uniquely mapped reads (UMR) was 20.2 M (10.8–31.68 M). The smallest detected aberration was a 1.12 Mb deletion, and the largest one represented a trisomy 16 (90 Mb) mosaic in a fetus (~ 20% on CMA). Detection sensitivity in the group with fetal fraction < 10% reached 57.14% (4/7). In the group with fetal fraction ≥ 10%, detection sensitivity was 100% (24/24) (Figure 2, Table 2).

The overall summary of the cohort and the basic characteristics of aberrations such as fetal fraction, size, and sequencing depth are summarized in Table 1.

The sex of the fetus was correctly determined in all cases. The median of differences between the CNV fraction and fetal fraction calculated from the Y chromosome or based on the “combined” method for each individual sample represented 4.9%, 5.0%, and 4.95% for XY, XX, and both sexes, respectively. Three cases were excluded from this “median of differences” calculation as outliers as the difference reached tens of percents. In 11 CNVs, the calculated CNV fractions per aberration indicated a maternal or a fetomaternal signal. The smallest aberrations represented a deletion and a duplication of 0.46 Mb, while the largest aberration was an 8.02 Mb duplication. The median CNV fraction calculated for maternal CNVs was 101.7% (SD ±4.78; range 96.5–109.9). These analyses were not included in the overall accuracy assessment, as it was not possible to distinguish whether or not they would have been detected in the event of exclusively fetal origin. Detailed information on maternal CNVs is provided in Table 2.

We recorded a case with a gain of 22q11.1–q11.21, known as causal for Cat eye syndrome (ORPHA 195, OMIM115470), mostly caused by partial tetrasomy of 22q11.1–q11.21, where the 22p marker chromosome was detected additionally by FISH and karyotyping.

## 4. Discussion

We performed a technical validation of cell-free DNA low-coverage whole-genome sequencing in combination with a CNV detection algorithm, which had performance in concordance with the CMA results. In total, we analyzed 34 plasma samples from high-risk pregnant women who underwent an invasive procedure (AMC) due to ultrasound findings or a family history comprising CNVs detected using CMA. Using our NIPT-based analysis, we recorded 40 notable CNVs. Out of these, 29 were recognized as fetal, and further 11 were categorized as maternal. Our data demonstrate that the detection sensitivity of our NIPT pipeline for CNVs reached 90.62% (29/32), regardless of the type, aberration size, fetal fraction, or resolution, including calls smaller than 3 Mb. Our observations are fully concordant with our previous work based on an in silico dataset and artificial DNA samples with a sensitivity of 79.3% for samples with ≥ 10% fetal fraction at 20 M reads, which further increased to 98.4% if only focused on deletions larger than 3 Mb [13]. To a great degree, this result can be compared with the 90.9% detection rate in a group of aberrations > 5 Mb. However, the performance was significantly worse in the group of < 5 Mb (14.3%), as reported by Li et al. This can probably be attributed to the lower read depth with a median 3.95 M UMR [21]. Three microdeletion cases were missed: 22q11.21(~ 2.6 Mb), 9p23 (~ 1.6 Mb), and 17p13.3 (~ 1.8 Mb). This was due to their small size (< 3 Mb) and low fetal fractions of 7.2, 8.3, and 8.36%, respectively. We detected a 22q13.32–q13.33 (~ 2.2 Mb) microduplication with a fetal fraction of 8.36%, indicating that detection limits with regards to fetal fraction and detectable size are close to 10% and 3 Mb. No false-positive CNVs were detected in our study; however, the number of samples was limited. Previous validations have demonstrated significant sensitivity for small CNVs on the order of hundreds of kilobases by increasing the sequencing depth to hundreds of millions of reads [9,22]. However, lower coverage sequencing can also deliver sufficient sensitivity: 78% for calls smaller than 3 Mb and even as much as 100% sensitivity for calls larger than 3 Mb. Very deep sequencing is not cost-effective or affordable in routine clinical settings, hence the threshold of 20 M UMR. This could be beneficial with respect to most clinically relevant CNVs, and it can undoubtedly increase clinical sensitivity. To address this issue, we performed an in silico downsampling to 20 M UMR on 17 analyses, which exceeded this limit (range 20.2–31.68). After downsampling, 2 of the 19 detected microaberrations previously detected at 28 M UMR and 31.61 M UMR were undetectable: 2.22 Mb 22q13.32–q13.33 (dup) and 2.86 Mb 22q11.22 (del), with fetal fractions of 8.36% and 10.8%, respectively. We compared a CNV fraction with fetal fraction based on Y mapping (for male fetuses) and the “combined” method (for female fetuses) and expected similar figures. The calculated CNV fraction was very close to the fetal fraction calculated from the Y chromosome or “combined” method. The median of their relative difference (i.e., relative_difference = abs × (ffCNV−ffY)/ffY) was only 4.9%, 5.0%, and 4.95% for XY, XX, and both sexes, respectively. The three outliers below were excluded from this comparison with a median of differences in tens of percents. The first was identified as a 1.12 Mb gain of 22q11.1–q11.21 with a fetal fraction of 10.8%. However, CNV fraction calculated for the given aberration was double (21.3%). Results from CMA, additional karyotyping, and FISH analysis indicated that the gain was due to tetraplication of 22q11.1–q11.21. The second represented a loss of 1.78 Mb of 22q11.21 with fetal fraction 9.3% versus 14.8% CNV fraction (an increase of 60%). This locus contained some unmappable regions and could, therefore, be a source for size- and CNV-fraction discrepancies (1.78 Mb WGS vs 2.66 Mb CMA). The third represented a 50% decrease from 11.2% fetal fraction to 5.4% for CNV fraction for the 13q33.2–q34 microdeletion. No definite explanation could be provided for this particular case. However, hidden biological reasons, such as a vanishing twin or confined placental mosaicism cannot be excluded.

As indicated above, it should be noted that our detection pipeline is capable of differentiating between a fetal CNV and a CNV from the mother (or the mother and the fetus). In a prospective setting, we noticed that very small aberrations are often calculated into the overall sensitivity [11,23] without evaluation of the z-score for the given aberration. We, therefore, assume that they would have been missed if they had not been maternal. We were able to detect 11 such maternal CNVs: eight microduplications and three microdeletions. Among them, a CMA assay from amniotic fluid confirmed a microduplication of Xp21.3 and a microdeletion of 20p12.3 as low as 0.46 Mb in a fetus. If these were purely fetal, they would have remained undetected with a fetal fraction of 11.28% and 16.1%, respectively.

There is a lack of pregnant plasma samples with CNVs available in a prospective setting to identify more precise detection limits. Based on a calculation from the data provided by Ke et al., the estimated frequency of reportable CNVs in the general pregnant population screened by massively parallel sequencing of pregnant plasma is 0.17% (24/14235) [5]. For illustrative purposes, we would have to analyze as a minimum a comparable number of samples expecting a pool of samples bearing CNVs similar to ours.

## 5. Conclusions

The results of our validation are concordant with the published data showing that sensitivity is a function of fetal fraction, size of CNV, sequencing depth, and other biology.

We successfully employed a reliable CNV-calling algorithm that detects aberrations larger than 3 Mb, a fetal fraction higher than 10% with a reasonable sequencing depth of 20 million uniquely mapped reads with sensitivity exceeding 90%. In addition, the calculation of the CNV fraction enabled differentiation between events of fetal and maternal origin.

Although our results are based on a limited number of samples and the dataset will need to be extended, there is evidence to suggest it will be feasible to translate this pipeline into a clinical screening setting.

## Figures and Tables

**Figure 1 diagnostics-10-00569-f001:**
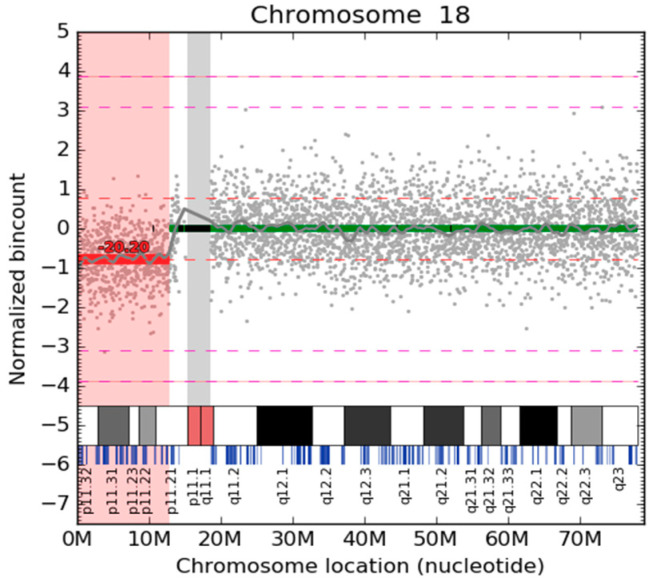
Visualization of CNV for chromosome 18. Normalized read counts per bin are depicted as gray dots. The red horizontal line shows the 18p11.32–p11.21 microdeletion approximately of 12.8 Mb. The light gray vertical band depicts an unmappable region around the centromere. Black horizontal bands signify bins that did not pass quality metrics and are thus excluded from the analysis. The light red region highlights the detected pathogenic region. The approximated *z*-score of deletion is displayed over the red segment. The estimated level of aberration detection based on the fetal fraction of this sample (12.8%) is visualized as a red dashed line, while the magenta dashed line represents the estimated level of maternal aberration detection.

**Figure 2 diagnostics-10-00569-f002:**
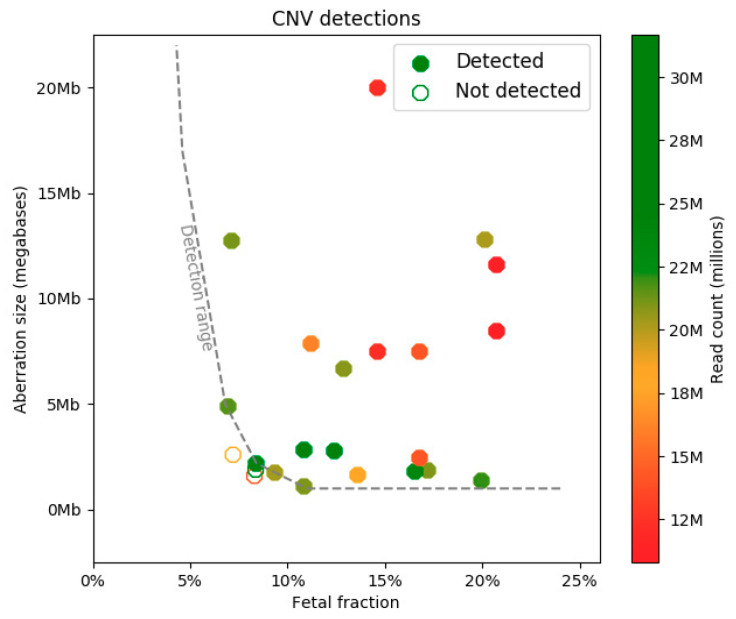
Detection sensitivity based on the fetal fraction and aberration size. Full markers depict the detected samples, while the empty ones represent the non-detected samples. The dashed gray line shows the estimated detection range.

**Table 1 diagnostics-10-00569-t001:** General characteristics of samples cohort.

	*N* = 34
**Maternal age (median)**	**31 (18–45)**
>35	27 (79.41%)
35–41	7 (20.58%)
**Gestational age in weeks (median)**	**17 (12–24)**
First trimester	3 (8.8%)
Second trimester	31 (91.17%)
**Weight (kg)**	**68 (51–87)**

**Table 2 diagnostics-10-00569-t002:** Evaluation of Copy Number Variation (CNV) detection.

According to CNV Size	Sensitivity (%)
Overall	90.62
< 3 Mb	78.57
≥ 3 Mb	100
**According to fetal fraction**	
< 10%	57.14
≥ 10%	100

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
