# Peer review of "Validation of Copy Number Variants Detection from Pregnant Plasma Using Low-Pass Whole-Genome Sequencing in Noninvasive Prenatal Testing-Like Settings"

_diagnostics, 2020, doi:10.3390/diagnostics10080569_

Round 1

Reviewer 1 Report

 This manuscript reports a study to validation of copy number variants detection from 2 pregnant plasma using low-pass whole-genome 3 sequencing in noninvasive prenatal testing.

I have some comments/questions for the abstract.

  1. The abstract and introduction is devoid of any real background to introduce the subject and provide the work with a context. I feel more background information is required.
  2. Abbreviation are present in the introduction and text and no definitiona are provided making it quite difficult to follow the study, e,g. `NGS` and `NIPT.`
  3.  Is this study solely validated for plasma and are there stability issue or particular sampling requirements that have to be adhered to for sampling accuracy? No clear information apparent.
  4. Table 2 was virtually impossible to read and requires amending for clarity.
  5. I feel another table is required to illustrate the actual findings of this study as some of the findings of the study were obscured in the text.
  6. The manuscript requires some details of the actual uniqueness of this study compared to previous studies and how the results of this study will actually benefit prenatal testing.
  7. In the conclusion the sentence is provided ` We have employed a reliable calling algorithm that detects the majority 217 of CNVs (> 3 Mb),` why is it unable to detect all  the CNVs, what is the limitation to the method?
  8.  Based on the results present, I don`t feel that the sentence `ranslation of this pipeline into a prospective screening setting is both feasible and beneficial` is justified and the conclusion should be toned down in accordance with the actual findings/limitations of the study together with the limited number samples investigated.

Author Response

  1. The abstract and introduction is devoid of any real background to introduce the subject and provide the work with a context. I feel more background information is required.

More information was added to introduction to point out the most relevant theses in the field of NIPT

  1. Abbreviation are present in the introduction and text and no definitiona are provided making it quite difficult to follow the study, e,g. `NGS` and `NIPT.

Definitions are provided (lines 32,34)

  1. Is this study solely validated for plasma and are there stability issue or particular sampling requirements that have to be adhered to for sampling accuracy? No clear information apparent.

The validation experiment was performed solely on plasma samples. All procedural steps and storage conditions regarding blood and plasma were standardized and mentioned in chapter Material and methods (Blood draw and plasma DNA isolation). We have added more detailed information about sample handling and quality checkpoint after the isolation of plasma DNA (lines 210-221). In general, the concentration of cf DNA in plasma is very low, often even immeasurable, and it is therefore important to avoid contamination of pregnant plasma with high molecular weight DNA originated from maternal nuclear cells.

  1. Table 2 was virtually impossible to read and requires amending for clarity.

Both tables (Table 1 and 2) with the most important findings were originally uploaded also as separate files with higher resolution. The tables are pasted only for the positioning in the manuscript and will be formatted and resized by editor. Regarding table format and type of data, we have inspired by previously published studies (e.g., reference 11) We incorporated as much information as possible to make them accessible for a later data mining (metaananalyses)

  1. I feel another table is required to illustrate the actual findings of this study as some of the findings of the study were obscured in the text.

We added another table (Table 3) summarizing important findings regarding sensitivity to CNV detection so it can be easily to follow at first look

  1. The manuscript requires some details of the actual uniqueness of this study compared to previous studies and how the results of this study will actually benefit prenatal testing.

We tried to highlight and communicate the most important issues of this validation in Discussion (e.g., differentiating between fetal and maternal genome, detection sensitivity for aberrations 3Mb, 10% fetal fraction, 20M UMR, CNV fraction)

  1. In the conclusion the sentence is provided ` We have employed a reliable calling algorithm that detects the majority 217 of CNVs (> 3 Mb),` why is it unable to detect all  the CNVs, what is the limitation to the method?

We tried to explain which factors and limitations are the most prominent for CNV detection (conclusion completely revised)

  1. Based on the results present, I don`t feel that the sentence `ranslation of this pipeline into a prospective screening setting is both feasible and beneficial` is justified and the conclusion should be toned down in accordance with the actual findings/limitations of the study together with the limited number samples investigated.

We modified and toned down our statement according to your suggestion, as the number of samples is really limited

Reviewer 2 Report

The authors extended their previous in silico findings as they reported in the text. Indeed, they performed a validation on clinical samples.

Minor revision:

  • keywords: you chose "Trisomy Test". I cannot find any mention of "Trisomy test" in the text. Were the partecipating women examined using this test ?
  • line 32: full name for NIPT as the first time you write it: xxxx (NIPT)
  • line 55: the same for NIPSCCD: full description and the (NIPSCCD)
  • line 65: "An aim of this validation ..." so you have other aims ? maybe you could rephrase

Author Response

  • keywords: you chose "Trisomy Test". I cannot find any mention of "Trisomy test" in the text. Were the partecipating women examined using this test ?We have decided to exclude the TRISOMY test from keywords as it is not used in the following text and is not essential for the content of the paper. 
  • Purely to supplement the information, the TRISOMY test is a commercial trademark of the best-selling non-invasive prenatal test in Slovakia and authors had participated in its implementation into clinical practice.
  •  
  • line 32: full name for NIPT as the first time you write it: xxxx (NIPT)
  • line 55: the same for NIPSCCD: full description and the (NIPSCCD)

We have added full name for NIPT and the same for NIPSCCD in a chapter where it is first time mentioned.

  • line 65: "An aim of this validation ..." so you have other aims ? maybe you could rephraseThe objective was to evaluate the clinical feasibility of NIPT to detect CNVs of different sizes at various fetal fractions on real pregnant plasma samples using CNV fraction based algorithm.
  • text was rephrased: